# A New qPCR Assay for the Rapid Diagnosis of *Anthonomus grandis* Subspecies

**DOI:** 10.3390/insects14110845

**Published:** 2023-10-31

**Authors:** Tyler Jay Raszick, Lindsey C. Perkin, Alejandra Godoy, Xanthe A. Shirley, Karen Wright, Paxton T. Martin, Charles P. -C. Suh, Raul Ruiz-Arce, Gregory A. Sword

**Affiliations:** 1Department of Entomology, Texas A&M University, College Station, TX 77843, USA; karen.wright@agr.wa.gov (K.W.); ptmartin97@gmail.com (P.T.M.); gasword@tamu.edu (G.A.S.); 2USDA-ARS Insect Control and Cotton Disease Research Unit, College Station, TX 77845, USA; lindsey.perkin@usda.gov (L.C.P.); charles.suh@usda.gov (C.P.-C.S.); 3USDA-APHIS-PPQ Science & Technology, Insect Management and Molecular Diagnostics Laboratory, Edinburg, TX 78541, USA; alejandra.godoy@usda.gov (A.G.); raul.a.ruiz@usda.gov (R.R.-A.); 4USDA-APHIS-PPQ, College Station, TX 77845, USA; xanthe.a.shirley@usda.gov

**Keywords:** boll weevil, thurberia weevil, molecular diagnostics, qPCR, TaqMan, SNP genotyping

## Abstract

**Simple Summary:**

Rapid diagnostic tools are critical for the management and eradication of boll weevil, *Anthonomus grandis grandis*. Here, we present the development and validation of a novel qPCR assay that enables same-day identification of *Anthonomus grandis* subspecies.

**Abstract:**

Rapid and accurate identification of *Anthonomus grandis* subspecies is crucial for effective management and eradication. Current diagnostic methods have limitations in terms of time to diagnosis (up to seven days) and can yield ambiguous results. Here, we present the validation of a custom TaqMan SNP Genotyping Assay for the rapid and accurate identification of *A. grandis grandis* (boll weevil) and *A. g. thurberiae* (thurberia weevil) subspecies. To validate the assay, we conducted three main experiments: (1) a sensitivity test to determine the DNA concentration range at which the assay performs, (2) a non-target specificity test to ensure no amplification in non-target weevils (false positives), and (3) an accuracy test comparing the results of the new assay to previously established methods. These experiments were carried out in parallel at three independent facilities to confirm the robustness of the assay to variations in equipment and personnel. We used DNA samples from various sources, including field-collected specimens, museum specimens, and previously isolated DNA. The assay demonstrated high sensitivity (PCR success with ≥0.05 ng/µL DNA template), specificity (0.02 false positive rate), and accuracy (97.7%) in diagnosing boll weevil and thurberia weevil subspecies. The entire workflow, including DNA extraction, assay preparation, PCR run time, and data analysis, can be completed within a single workday (7–9 h) by a single technician. The deployment of this assay as a diagnostic tool could benefit boll weevil management and eradication programs by enabling same-day diagnosis of trap-captured or intercepted weevil specimens. Furthermore, it offers a more reliable method for identifying unknown specimens, contributing to the overall effectiveness of boll weevil research and control efforts.

## 1. Introduction

The boll weevil, *Anthonomus grandis grandis* Boheman (Coleoptera: Curculionidae), is a historically important economic pest of commercially cultivated upland cotton, *Gossypium hirsutum* L. (Malvales: Malvaceae), in the United States. From its introduction and subsequent range expansion across the US Cotton Belt in the early 20th century, the boll weevil devastated the US cotton industry until a nationwide eradication program was implemented in the latter half of the century [1,2,3]. The US Boll Weevil Eradication Program has been widely successful, eliminating the species from nearly all of the country’s cotton-growing regions, but it continues to impact cotton production in the Lower Rio Grande Valley (LRGV) of southern Texas, and it is endemic to the bordering region of northern Tamaulipas, Mexico [3]. Despite coordinated international efforts to eradicate the boll weevil in the LRGV and northern Tamaulipas, complete eradication remains elusive due to the large size of a population that is contiguous across international borders, a climate that is favorable for weevil development, and a year-round availability of volunteer cotton and other suitable hosts that potentially support boll weevil reproduction and survival [4,5,6,7,8]. Nonetheless, eradication efforts continue in order to prevent the establishment of infestations in local cotton and to maintain an effective quarantine [3].

Management of the boll weevil is further complicated by the existence of a morphologically similar subspecies, the thurberia weevil (*A. g. thurberiae*), which is distinguished by its host plant preference for the wild cotton, *Gossypium thurberi* Tod. (Malvales: Malvaceae) [1,4,9,10]. Typically, the thurberia weevil has been regarded as a non-pest variant, though it should be noted that multiple recent genetic investigations have suggested that the two subspecies might be more accurately described as geographic lineages that may include both pest and non-pest populations [8,9,11]. Boll weevil management and eradication programs rely on pheromone-baited cone traps for monitoring and early detection of incipient populations [12,13,14,15]. When one or more suspect weevils are captured in a trap, programs increase the trap density around the area and execute a series of insecticidal sprays in nearby cotton fields to prevent establishment of a pest population. However, both the pest and the non-pest variants of *A. grandis* respond to the pheromone bait, are captured in traps, and trigger the management response [16]. In addition to trapping, *Anthonomus* spp. weevils are also often intercepted at US ports of entry. An ad hoc search of the AQAS database returned 4005 such interceptions from 1984 to 2021.

Contemporary diagnostic methods used by the US Department of Agriculture Animal and Plant Health Inspection Service (USDA-APHIS) to distinguish between *A. g. grandis* and *A. g. thurberiae* include classical taxonomy (especially a morphometric ratio of the profemora) and a cytochrome *c* oxidase subunit I (COI) sequencing assay developed by Barr et al. [17]. The morphometric measurement is limited in its utility because there is a range of ratios for which the determination is ambiguous, and the morphological characteristics used for the measurement may be labile to diet content or quality [1,10,18]. The COI assay method is reported to be approximately 94% accurate in distinguishing between the subspecies, but it can take up to five days for a determination [17]. Finally, weevil samples that are in poor condition, as those recovered from pheromone traps often are, can make determinations difficult by either method. From 2015 to 2021, 1214 suspect trap-captures were analyzed at the USDA-APHIS Insect Management and Molecular Diagnostics Laboratory (Table 1), and 11% of the trap-captures could not be effectively identified using existing methods. Whether weevils are captured in traps or intercepted at ports of entry, rapid and accurate identification of suspect weevil samples is critical to the success of management and eradication programs and to facilitate uninterrupted international trade. Determinations failing to recognize the pest variant could lead to the establishment of pestiferous boll weevil populations, which are costly to mitigate. Conversely, if thurberia weevil is incorrectly identified as boll weevil, then there could be losses in revenue due to unnecessary and costly pesticide applications. Incorrect identifications can also trigger unnecessary and costly quarantines. Thus, there is a need for a reliable assay that can consistently yield correct diagnoses in a manner that is rapid enough to provide timely recommendations to management.

Advancements in diagnostic technologies have enabled the development of rapid workflows that leverage qualitative real-time polymerase chain reaction (PCR) methods to detect fixed variants of single-nucleotide polymorphism (SNP) loci that are diagnostic at the subspecific level [19]. SNP-based PCR diagnostic methods have been successfully developed for a wide variety of other species [19,20]. Here, we present the development and validation of a custom TaqMan^®^ SNP Genotyping Assay that can rapidly and accurately diagnose *A. g. grandis* and *A. g. thurberiae*.

## 2. Materials and Methods

### 2.1. Initial Assay Design and Locus Selection

We screened a large, previously generated SNP dataset for candidate loci that could be used to reliably diagnose boll weevil and thurberia weevil variants [8]. Argentine boll weevil specimens from those data were omitted from this analysis because they were considered unlikely to be a potential source of trap-captures or intercepts in North America. Additionally, specimens obtained from traps near commercial cotton in Sonora, Mexico, were considered to be thurberia weevil, consistent with the findings of previous genetic studies [8,9,11]. We also included specimens from 2 weevil trap-capture events: 24 weevils trapped along the US–Mexico border in Hidalgo Co., NM, USA, in 2017, and 56 weevils from a 2015–2018 re-infestation of the previously eradicated area in Winter Garden, TX, USA [21]. Ultimately, 193 boll weevils representing 3 genetic populations and 131 thurberia weevils representing 2 genetic populations were evaluated. To find the best candidate loci for assay development, we calculated the F_ST_ values between the variants at every biallelic SNP locus using RStudio v1.1.456 and the package R/genepop v1.1.4 [22,23,24]. An ideal locus for diagnostic assay development should be alternatively fixed in each weevil variant—one allele should occur only in the homozygous condition in *A. g. grandis* and the alternate allele should occur only in the homozygous condition in *A. g. thurberiae*. Such a locus would have an F_ST_ value of 1 when calculating the pairwise F_ST_ between the two subspecies, indicating complete alternative fixation. No locus yielded an F_ST_ value of 1, so we selected the top ten candidate loci (F_ST_ > 0.94) for initial testing. We also checked the genotype counts for each locus across the dataset to ensure that the heterozygous condition was rare and that there were no alternative homozygotes in either subspecies. Next, we located the candidate loci in the preliminary genome sequence and extracted them along with 500 base pair flanking regions [8]. These 1001 base pair sequences were then submitted to the Custom TaqMan^®^ Assay Design Tool (https://www.thermofisher.com/order/custom-genomic-pro–ducts/tools/cadt/, accessed on 14 June 2019). Upon design and delivery of the ten candidate assays, we screened their performance using DNA samples from vouchered specimens previously identified with both morphological and molecular tools (data available upon request). Each assay’s performance was evaluated with regards to amplification efficiency and the presence/absence of undesirable heterozygote calls. After multiple rounds of screening with increasingly greater sample sizes, we determined that one assay, ID: ANZTMGE (Table 2), was suitable for further validation.

### 2.2. Validation of the ANZTMGE Assay

#### 2.2.1. Experimental Approach to Assay Validation

The approach for the validation of the ANZTMGE assay consisted of three main experiments: (1) a sensitivity test to determine the acceptable range of DNA concentrations at which the assay could be expected to amplify efficiently, (2) a non-target specificity test to ensure that the assay would not amplify in non-target weevils that might be encountered in boll weevil traps (and thereby yield false positives or amplifications from non-target species, leading to incorrect determinations), and (3) an accuracy test to determine if the assay yielded results consistent with existing assays. This accuracy test also doubled as a test of the assay’s performance on samples sourced from across a broad geographic range.

All three validation experiments were carried out in parallel at three different laboratory facilities with different equipment and personnel to confirm that the results were reproducible under variation in equipment and with different individuals performing the benchwork. Specifically, experiments were carried out at the USDA-APHIS Insect Management and Molecular Diagnostic Laboratory in Edinburg, Texas (IMMDL), the USDA-ARS Southern Plains Agricultural Research Center, Insect Control and Cotton Disease Research Unit in College Station, Texas (ICCDRU), and the Entomology Research Laboratory Building at Texas A&M University in College Station, Texas (TAMU).

#### 2.2.2. Specimen Sampling and DNA Isolation

The boll weevil DNA samples included in each validation experiment varied according to the objective being addressed by each test. For the sensitivity test, we obtained previously identified boll weevil specimens from the USDA-APHIS collection maintained at IMMDL. This collection consisted of field-collected weevils and weevils that had been reared in laboratory colonies. For the non-target specificity test, we used field-collected samples and obtained museum specimens from the Texas A&M University Insect Collection (TAMUIC). Only museum specimens that were initially identified by widely recognized experts with extensive background and contributions to scientific literature on systematics and taxonomy of New World weevils were selected and used in the sensitivity test. The collection dates across the TAMUIC specimens used ranged from 1937 to 2009. For the accuracy test, we obtained previously isolated DNA from a subset of specimens that were used for the COI assay development [17]. Prior to DNA isolation, TAMUIC specimens were pinned or point-mounted and stored at room temperature, and all other samples were stored at −20 °C.

For those weevil samples that were not already in the form of eluted DNA, we carried out DNA isolation using one of two different methods depending on the condition of the sample. Both methods utilized a Qiagen DNA isolation kit and followed the manufacturer’s standard protocol for animal tissues. For fresh and ethanol-fixed specimens, the Qiagen DNeasy Blood and Tissue Kit (Qiagen, Hilden, Germany) was used. Weevils were mechanically disrupted in the tissue lysis buffer (whole body or leg) with a laboratory pestle prior to the addition of proteinase K. For museum specimens, DNA was isolated using the Qiagen QIAamp kit (Qiagen, Hilden, Germany) and a nondestructive method wherein individuals were submerged overnight in the lysis buffer with proteinase K and gentle mixing. Carrier RNA was used to improve the binding of DNA to the elution column membrane, as is recommended for potentially fragmented DNA in low concentrations, such as museum specimens. The carrier RNA was provided in the Qiagen QIAamp DNA extraction kit and used at a final concentration of 1 µg/µL. RNase A was used to remove the carrier RNA before the elution of DNA. The concentrations of the eluted DNA samples were quantified with a Qubit Flex instrument (Thermo Fisher Scientific Inc., Waltham, MA, USA) using high-sensitivity dsDNA kits (Appendix A). To avoid unnecessary freeze–thaw cycles, isolated DNA samples were stored at 4 °C if they were to be used for an experiment in less than one week. Otherwise, samples were stored at −20 °C.

We performed an additional six DNA isolations from three previously identified boll weevils and three previously identified thurberia weevils to serve as positive control (+C) groups. We extracted and isolated DNA from the entire weevil to maximize the total yield of DNA. The DNA was extracted and isolated from the +C weevils using the same methods as for fresh and ethanol-fixed specimens.

#### 2.2.3. PCR Setup and Run Conditions at IMMDL and ICCDRU

All amplifications were performed on a Bio-Rad CFX96 Touch^TM^ Real-time PCR Detection System (Bio-Rad, Hercules, CA, USA) following the manufacturer’s protocol. Each 10 µL reaction contained 4.5 µL of DNA template (0.05–400 ng/µL, depending on the test), 5.0 µL of 2× TaqMan Genotyping Master Mix, and 0.5 µL of 20× TaqMan assay probes. PCR cycle conditions were as follows: 95 °C for 10 min, 95 °C for 15 s, 60 °C for 1 min × 45, 25 °C for 30 s, and 4 °C ∞. Triplicate reactions were performed for each individual sample, except in the accuracy tests, which were performed only in duplicate due to limited voucher DNA. Each plate was set up to include six +C samples (three *A. g. grandis* and three *A. g. thurberiae*) and three no-template controls (NTC), all in triplicate or duplicate, as the sample volume allowed.

#### 2.2.4. PCR Setup and Run Conditions at TAMU

All amplifications were performed on a Bio-Rad CFX384 Touch^TM^ Real-Time PCR Detection System (Bio-Rad, Hercules, CA, USA). Amplifications were performed in 5 µL reactions instead of 10 µL, as at IMMDL and ICCDRU. Nonetheless, the ratio of reagents remained consistent. Each reaction contained 2.25 µL of DNA template, 2.50 µL of 2× TaqMan Genotyping Master Mix, and 0.25 µL of 20× TaqMan assay probes. Cycle conditions also remained the same as at the other facilities, and again, each plate included three *A. g. grandis* +C, three *A. g. thurberiae* +C, and three NTC, all in triplicate or duplicate, as the sample volume allowed. 

#### 2.2.5. Quantification Cycle and Endpoint Analysis and Interpretation for Variant Diagnosis

Completed PCR amplifications at all three facilities were collected and analyzed using Bio-Rad CFX Maestro^TM^ 2.0 v5.0.021.0616 software. The quantification cycle (Cq) values (the PCR cycle number at which fluorescence is detected above a background threshold) were obtained for both fluorophores for every reaction. Where amplification was successful, the mean Cq value was calculated for both fluorophores across all reactions in the sensitivity and accuracy experiments. The non-target specificity test was excluded from these calculations because amplification was not expected for those samples.

For reactions in which only one fluorophore was detected, the reaction was assigned as indicative of homozygosity for either the boll weevil allele or the thurberia weevil allele, depending on which fluorophore was detected. For reactions in which both fluorophores were detected, we calculated the absolute difference in Cq values between the fluorophores (ΔCq) and then calculated the mean ΔCq for both homozygous genotypes. 

“Endpoint analysis” refers to the analysis of fluorescence data for the final cycle of the PCR run (Figure 1). We used CFX Maestro to analyze the fluorescence data for the final cycle of the PCR run and to make “calls” regarding the genotypes in each reaction. Calls could indicate: (1) homozygosity for the FAM-labeled allele (the thurberia weevil allele), (2) homozygosity for the VIC-labeled allele (the boll weevil allele), (3) heterozygosity, or (4) “no call.” The detection of heterozygosity was expected to be rare because the two alleles are nearly alternatively fixed in the two variants of the species. A “no call,” usually indicative of a failed PCR, was expected for the NTCs but could have also occurred due to human or equipment error or possible false negatives.

For any given individual sample, the final variant determinations were performed at each participating facility by first reviewing the results of the endpoint analysis performed at the respective facility and applying the following criteria: (1) if all replicate calls agreed, the final determination was the same as the calls; (2) if replicates did not agree, the amplification curves for the replicates were examined for an exponential pattern of amplification, which is expected for a normal PCR. If any curves did not exhibit such a pattern, those replicates were considered defective (likely resulting from human and/or equipment errors) and discarded. If amplification curves appeared to indicate proper amplification, conflicting determinations among replicates were resolved using the following criteria: (a) if there was a single heterozygote or “no call” determination and the other replicates agreed with each other, the replicates in agreement were regarded as correct and the replicate indicating otherwise was ignored; (b) if there were multiple heterozygote or “no call” replicates, or if all replicates differed from one another, the final call was categorized as “inconclusive.” The complete stepwise determination process is visualized in Figure 2.

#### 2.2.6. Sensitivity Test

The integrity and quantity of DNA extracted from dead trap-captured boll weevils may considerably decline over time, so the ANZTMGE assay’s suitability for amplification was tested across a field-relevant range of DNA concentrations (0.02–36.6 ng/µL) [25]. DNA isolated from both weevil variants (N = 5 for both variants) was serially diluted to the desired concentrations. First, each DNA sample was diluted to approximately 25 ng/µL (ranged between 22.9 ng/µL and 26.5 ng/µL). Next, samples were serially diluted to achieve target concentrations of 5 ng/µL, 1 ng/µL, and 0.05 ng/µL (Table 3). These concentrations reflected the expected yield of DNA isolations carried out on single legs of trap-captured specimens [25]. Therefore, this experiment also tested the ANZTMGE assay’s ability to perform as desired when sample DNA is either limited or in poor condition, such as may be the case for dead weevils that have been in traps for extended periods of time [25].

#### 2.2.7. Non-target Specificity Test

The ANZTMGE assay was challenged using closely related species within the genus *Anthonomus* and other weevil species occasionally encountered in boll weevil traps. This experiment tested the assay’s robustness against the possibility of false positive determinations. For this test, we utilized 34 museum specimens on loan from TAMUIC and 21 fresh or ethanol-fixed specimens from IMMDL and ICCDRU field collections. DNA concentrations from these samples varied between 4.42 and 103 ng/µL (Appendix A). We included eight expert-identified (H.R. Burke and R.W. Jones) *Anthonomus* species (*A. eugenii*, *A. fulvus*, *A. hunteri*, *A. musculus*, *A. palmeri*, *A. peninsularis*, *A. texanus*, and *A. townsendi*) and three more distantly related species (*Conotrachelus nenuphar*, *Curculio caryae*, and *Rhyssomatus lineaticollis*) that co-occur in cotton-growing regions (Table 4). The latter two species may be captured in boll weevil traps when conditions promote their co-occurrence with *A. grandis* [16]. 

#### 2.2.8. Accuracy Test

DNA samples from 77 *A. g. grandis* and 52 *A. g. thurberiae* specimens that were previously used in the development and testing of the COI assay (Table 5) were used to evaluate the accuracy of the ANZTMGE assay [17]. Voucher DNA eluates were divided into three aliquots and one aliquot was sent to each participating facility where the ANZTMGE assays and endpoint analyses were subsequently carried out. The accuracy of the ANZTMGE assay was evaluated by assessing whether the final determinations made based on the ANZTMGE assay matched the previous determinations.

This experiment was also used to evaluate the performance of the ANZTMGE assay on specimens from across a broad geographic sampling range (Table 6). The 77 *A. g. grandis* specimens included representatives from eight sites in Mexico and three sites in the US, and the 52 *A. g. thurberiae* specimens included representatives from one site in Mexico and six sites in the US. Thus, this experiment allowed us to simultaneously test the accuracy of the ANZTMGE assay and evaluate its performance across geographically different weevil populations.

## 3. Results

### 3.1. Quantification Cycle Analysis

The mean ± s. d. Cq for the FAM-labelled allele (indicating thurberia weevil) was 29.93 ± 5.61, and 30.10 ± 6.14 for the VIC-labelled allele (indicating boll weevil). The CFX Maestro software was able to reliably make calls in the endpoint analyses, indicating a “no call” for only 8 out of 414 reactions in the sensitivity test (1.9%) and 125 of 874 reactions in the accuracy test (14.3%). For reactions in which the endpoint analysis indicated homozygosity and both fluorophores were detected, the mean ± s. d. ΔCq was 5.59 ± 2.73 for the thurberia weevil and 6.08 ± 4.52 for the boll weevil. Across all participating facilities and experiments, NTCs failed to amplify as expected.

### 3.2. Sensitivity Test

Results showed that the ANZTMGE assay performed as desired for a wide range of template DNA concentrations, though efficient qPCR amplification was found to be slightly more consistent for boll weevil samples (mean ± s. d. relative fluorescence units (RFUs) 7203 ± 3830) than for thurberia weevil samples (mean ± s. d. RFUs 6135 ± 5492). A total of 360 qPCR reactions (180 for boll weevil and 180 for thurberia weevil) were carried out across the 3 participating facilities. Among all boll weevil samples, only two reactions yielded a “no call” (99% PCR success rate). The two “no calls” occurred at different concentrations (1 ng/µL and 25 ng/µL) and at different facilities (one at IMMDL and one at TAMU). In both cases, the replicate reactions performed as expected, indicating that these “no calls” were likely due to human or equipment error. For thurberia weevil, nine reactions yielded calls inconsistent with expectations (95% PCR success rate). Of these, four of the failures consisted of two pairs of replicates at a single facility (IMMDL) that yielded heterozygote calls (both with DNA concentrations of 1 ng/µL). In all five of the other cases, the inconsistent calls were not corroborated by the other two replicates, and they occurred at DNA concentrations of 5 ng/µL or 25 ng/µL. Overall, there was no consistent pattern that indicated that the ANZTMGE assay might not perform effectively if appropriate concentrations of DNA from weevils are used. In fact, our results indicated that template DNA concentrations as low as 0.05 ng/µL would be acceptable for use with this assay, and users can expect a >95% PCR success rate when template DNA concentrations are above that level.

### 3.3. Non-Target Specificity Test

The ANZTMGE assay was found to be robust against possible false positive identifications due to amplification of DNA from non-target weevil species. For this test, amplification failure (of non-target weevil DNA) was desirable and expected. Of the 495 reactions (55 non-target individuals run in triplicate at 3 facilities; Table 3), 466 failed to amplify (6% PCR success rate). Of the 29 successful qPCR amplifications, 18 were due to the accidental inclusion of 2 previously misidentified specimens. Specifically, two TAMUIC specimens that had been previously identified and databased as *A. peninsularis* were diagnosed as *A. g. thurberiae* using the ANZTMGE assay. This result was consistent across all replicates and at all facilities. The previous misidentification was confirmed by reevaluating previous COI sequencing performed for those specimens and re-identification using traditional taxonomic methods. This event demonstrated the ability of the ANZTMGE assay to reliably detect *A. grandis* variants, even when the user may not be expecting such a result. Conversely, the otherwise low qPCR success rates across all other samples demonstrated the robustness of the assay against false positives, indicating that misclassification of other weevil species as either boll weevil or thurberia weevil will be an extremely rare occurrence when utilizing this assay.

### 3.4. Accuracy Test

Based on a subset of weevils from Barr et al., this experiment evaluated both the accuracy of the ANZTMGE assay and its robustness against geographic diversity in sample origin [17]. Overall, the ANZTMGE assay was found to be highly accurate: 75 of the 77 boll weevil samples (97.4% accuracy) and 51 of the 52 thurberia weevil samples (98.1% accuracy) were correctly identified, yielding an overall accuracy level of 97.7% (Table 7).

With regards to the assay’s performance across weevil geography, the assay was also found to be very robust (97.7% accuracy; Table 8). The two inconsistent determinations for boll weevil were both in samples from the LRGV region of Texas. These individuals had different COI haplotypes and only yielded inconsistent results at one of the three facilities, while the results at the other two facilities were consistent with the COI assay. Considering the haplotypic difference and the fact that the other facilities did not find this inconsistency, we are confident that there is a very low risk of assay failure for weevils obtained from the LRGV region. For thurberia weevil, the one sample yielding an inconsistent determination showed a haplotype designated as AN4 by Barr et al. [17]. The AN4 haplotype is commonly found in both the boll weevil and thurberia weevil, but based on the capture location and host plant, this specimen was most likely thurberia weevil. As with the two inconsistent boll weevil samples, only one facility yielded an inconsistent result for this individual.

## 4. Discussion

In this paper, we described and validated a new rapid genetic assay, denoted ANZTMGE, that enables cost-efficient subspecific diagnosis of boll weevil (*A. g. grandis*) and thurberia weevil (*A. g. thurberiae*) with 98% accuracy. The deployment of this assay as a diagnostic tool could directly benefit US cotton programs by enabling same-day diagnosis of trap-captured or intercepted putative cotton boll weevil specimens, and it may also benefit other boll weevil researchers by allowing for more rapid diagnoses of unknown specimens.

The validation experiments provided strong support that the ANZTMGE assay is highly reliable with regards to amplification success (>95% PCR success rate) and accuracy (97.7% accuracy), although it should be noted that despite the high accuracy of the assay, the overall qPCR success was lower in the accuracy experiment than in the sensitivity test (86.5% compared to 98.6%). These amplification failures led to a higher number of inconclusive final determinations in the accuracy experiment, which ultimately prevented the assay from being 100% accurate. However, considering the success of the sensitivity experiment, which showed that the assay performed well with very low amounts of DNA, we speculate that some other issue, such as DNA degradation, buffer evaporation, or the presence of PCR-inhibiting contamination, affected the downstream amplification performance (DNA concentration values for the samples can be found in Appendix A). Future assay development should include an internal PCR control to help explain anomalies in amplification performance and detect false negatives [26].

We did not observe any issues with the assay’s performance that could have been associated with the broad geographical origin of weevils. Our results support the assertion that the assay will perform well for a wide range of field-relevant DNA concentrations (<0.02–36.6 ng/µL) [25]. Additionally, the risk of the ANZTMGE assay producing false positives is extremely low—the assay successfully excluded 11 non-target weevil species that can be captured in boll weevil pheromone traps and/or have a similar appearance to the boll weevil. This allows for confidence when using the assay on closely related species or when specimens are too degraded for identification using morphological characters. Finally, this tool alleviates the issue of non-diagnostic mitochondrial haplotypes. As mentioned above (Section 3.4), Barr et al. reported the presence of an ambiguous mitochondrial haplotype that was observed to be common among both *A. g. grandis* and *A. g. thurberiae* collections, preventing any confident diagnosis of a sample with that haplotype [17]. The ANZTMGE assay uses nuclear diploid markers that are nearly alternatively fixed, so individuals with the AN4 haplotype, or any ambiguous mitochondrial haplotype, can be readily diagnosed.

If deployed, this tool could significantly reduce the time and cost required to accurately determine the identity of a suspect weevil, which is critical for boll weevil eradication programs. This new assay can provide diagnoses within eight working hours, a substantial improvement over the five working days required for the COI assay. The reagent cost per sample is roughly USD 6.00, as opposed to USD 12.00 for the COI assay. Finally, the benchwork can be completed in a single day by a single technician, whereas the COI assay requires multiple days of benchwork. Therefore, costs are further lowered by reducing the man-hours required to run the assay.

Protocols for running the assay and analyzing run data using CFX Maestro software have been made available in Appendix A, but other software may be used to analyze fluorescence data from qPCR. Users wishing to analyze qPCR data with other software should refer to the Cq values presented in this paper to establish baseline minimums for fluorophore detection. It is highly recommended that assays be run in triplicate to provide the highest confidence in the diagnosis. The assay is already commercially available and can be obtained from Thermo Fisher Scientific by ordering a Custom TaqMan SNP Genotyping Assay using the Reorder Existing Custom Assays option and the assay ID name ANZTMGE.

## Figures and Tables

**Figure 1 insects-14-00845-f001:**
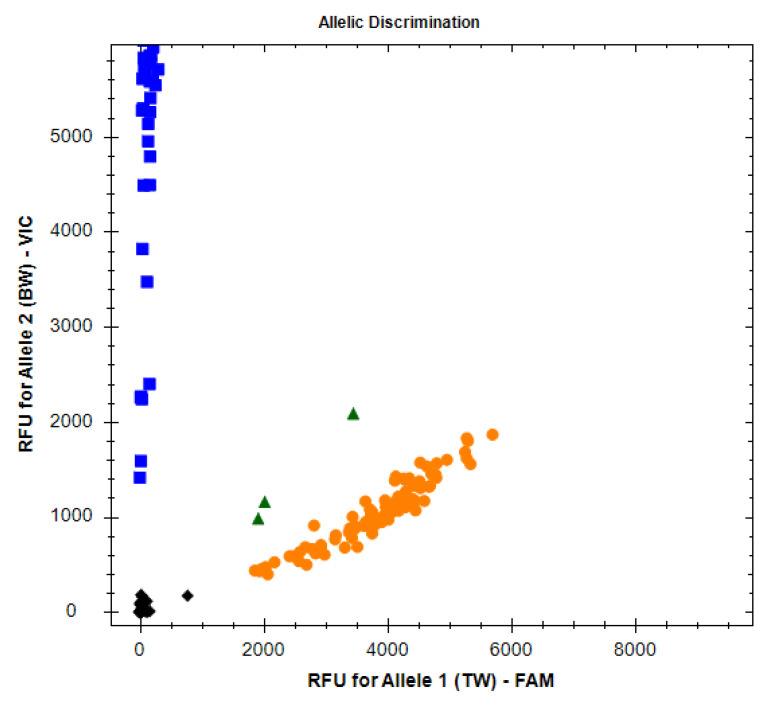
Example of CFX Maestro endpoint analysis indicating the RFU values for samples that were amplified as homozygous for the VIC-labeled BW allele (blue squares), homozygous for the FAM-labeled TW allele (yellow circles), or heterozygous (green triangles). No-template controls and samples for which amplification failed are shown as black diamonds. This example is the result of the TAMU facility’s accuracy test.

**Figure 2 insects-14-00845-f002:**
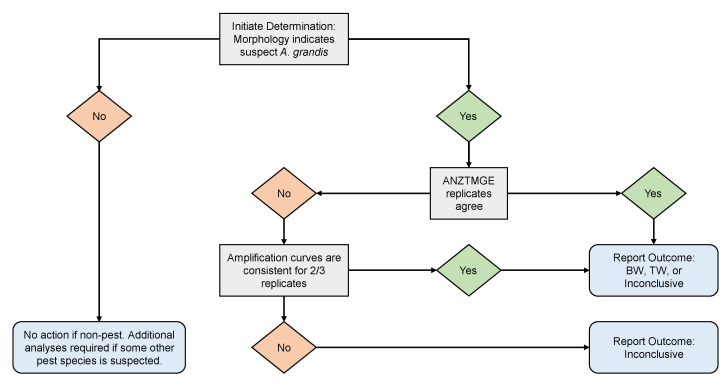
Decision process flowchart describing the criteria used for *Anthonomus grandis* subspecies determinations based on results from the ANZTMGE assay.

**Table 1 insects-14-00845-t001:** Identifications of suspect *Anthonomus grandis* weevils carried out at the USDA-APHIS Insect Management and Molecular Diagnostics Laboratory from 2015 to 2021.

Total Captures	*A. g. grandis*	*A. g. thurberiae*	Undet. *A. grandis*	Non-*A. grandis*
1214	33	1036	134	11
	3%	85%	11%	1%

**Table 2 insects-14-00845-t002:** Primer, probe, and PCR product information for the ANZTMGE assay. Reporter 1 dye is VIC and reports the boll weevil (*A. g. grandis*) allele. Reporter 2 dye is FAM and reports the thurberia weevil (*A. g. thurberiae*) allele.

Forward Primer	5′-CTGGCACTGTCGCGAATCTAT-3′
Reverse Primer	5′-ACGGACCGTTAGAAAAATACTTGGT-3′
Reporter 1 Sequence	5′-AAGCCGATCTTGCTAGTT-3′
Reporter 2 Sequence	5′-AAAGCCGATCTTACTAGTT-3′
Context Sequence	5′-GGCACTGTCGCGAATCTATAACTAG[C/T]AAGATCGGCTTTACCAAGTATTTTT-3′

**Table 3 insects-14-00845-t003:** List of *A. g. grandis* and *A. g. thurberiae* specimens used in examining the sensitivity of the ANZTMGE assay. Target DNA concentrations and the actual range of concentrations used are listed. The quantity (N) of samples examined is also shown.

Subspecies	Locality	N	Target (DNA) (ng/µL)	Actual Range (DNA) (ng/µL)
*A. g. grandis*				
	USA, Texas, Knippa	3	0.05	0.029–0.252
	USA, Texas La Feria	2	0.05	0.018–0.172
	USA, Texas, Knippa	3	1	0.064–0.901
	USA, Texas, La Feria	2	1	0.538–0.828
	USA, Texas, Knippa	3	5	3.32–6.92
	USA, Texas, La Feria	2	5	2.76–5.94
	USA, Texas, Knippa	3	25	21.8–36.6
	USA, Texas, La Feria	2	25	16.6–32.2
*A. g. thurberiae*				
	USA, New Mexico	5	0.05	0.01–0.127
	USA, New Mexico	5	1	0.55–1.68
	USA, New Mexico	5	5	2.46–7.18
	USA, New Mexico	5	25	11.7–32.2

**Table 4 insects-14-00845-t004:** Species and origin of weevil specimens (N = 55) used in the ANZTMGE assay to examine non-target specificity. “n/a” indicates that there is no common name approved and accepted by the Entomological Society of America.

Species	Common Name	Origin	N
*A. eugenii*	pepper weevil	USA, Florida	3
		USA, Texas	2
*A. fulvus*	winecup weevil	USA, Texas	5
*A. hunteri*	n/a	Mexico, Campeche	5
*A. musculus*	cranberry weevil	USA, New Jersey	4
*A. palmeri*	n/a	Mexico, Chiapas	5
*A. peninsularis*	n/a	USA, Arizona	2
		USA, California	4
*A. texanus*	n/a	Mexico, Chihuahua	2
		USA, Texas	3
*A. townsendi*	n/a	Mexico, Chiapas	5
*Conotrachelus nenuphar*	plum curculio	USA, Oklahoma	2
		USA, Vermont	2
		USA, Texas	1
*Curculio caryae*	pecan weevil	USA, Texas	5
*Rhyssomatus lineaticollis*	milkweed stem weevil	USA, Wisconsin	2
		USA, Texas	2
		USA, Missouri	1

**Table 5 insects-14-00845-t005:** Subspecies, geographical locality, and COI haplotype assignments for a subset of *Anthonomus grandis* specimens used in Barr et al. [17] to evaluate the accuracy of the ANZTMGE assay. N is the number of weevil specimens for each combination of locality and haplotype.

Subspecies		Country	Locality	Haplotype
*A. g. grandis*				
	Mexico			
		Chihuahua	-	2
		Chihuahua	AN4	2
		Chihuahua	AN11	1
		Chihuahua	AN12	25
		Coahuila	AN4	2
		Coahuila	AN12	4
		Coahuila	AN25	1
		Durango	AN4	1
		Durango	AN12	6
		Tamaulipas	-	2
		Tamaulipas	AN1	5
		Tamaulipas	AN28	2
		Tamaulipas	AN30	1
	USA			
		Mississippi	AN1	2
		Mississippi	AN9	1
		Mississippi	AN13	2
		New Mexico	AN1	1
		Texas	AN9	1
		Texas	AN10	1
		Texas	AN13	4
		Texas	AN27	7
		Texas	AN28	3
		Texas	AN31	1
*A. g. thurberiae*				
	Mexico			
		Chihuahua	-	1
		Chihuahua	AN4	2
		Chihuahua	AN8	1
		Chihuahua	AN14	2
		Chihuahua	AN15	1
		Chihuahua	AN16	4
		Chihuahua	AN17	1
		Chihuahua	AN18	1
		Chihuahua	AN19	2
		Chihuahua	AN20	1
		Chihuahua	AN21	1
	USA			
		Arizona	AN2	2
		Arizona	AN4	1
		Arizona	AN8	3
		Arizona	AN17	8
		Arizona	AN18	2
		Arizona	AN19	10
		Arizona	AN20	3
		Arizona	AN22	1
		Arizona	AN23	1
		Arizona	AN24	2
		Arizona	AN26	2

**Table 6 insects-14-00845-t006:** Geographic locality information for the 129 weevil specimens used in Barr et al. [17] that were also used to evaluate the robustness of the ANZTMGE assay to geographical variations in weevil origin. N is the number of weevil specimens from each locality.

Subspecies	Country	Origin	N
*A. g. grandis*			
	Mexico		
		Chihuahua, Delicias	13
		Chihuahua, Durango, Torreon	5
		Chihuahua, Ojinaga	8
		Chihuahua, Los Alamos	4
		Coahuila, Ejido Chavez	5
		Coahuila, San Pedro	2
		Durango, Jimenez	7
		Tamaulipas, Valle Hermoso	10
	USA		
		Mississippi, Meyersville	5
		New Mexico, Artesia	1
		Texas, Rio Grande Valley	17
*A. g. thurberiae*			
	Mexico		
		Chihuahua, Agua Prieta	17
	USA		
		Arizona, Bisbee	1
		Arizona, East of Sasabe	12
		Arizona, Kitt Peak	9
		Arizona, Phoenix	5
		Arizona, Sonoita	4
		Arizona, Chiricahua, Mts.	4

**Table 7 insects-14-00845-t007:** Diagnostic identifications of a subset of 129 weevil specimens obtained from the Barr et al. [17] voucher specimens using the ANZTMGE assay. From left to right, the columns correspond to the quantity of specimens per collection (N), weevil identity determinations (BW = boll weevil, TW = thurberia weevil, Inc. = inconclusive) made at each facility (IMMDL, TAMU, and ICCDRU) using the stepwise criteria (Figure 2), the PCR success rate for each collection, and the accuracy of the ANZTMGE assay for each collection. Determinations inconsistent with Barr et al. (2013) are bolded and underlined.

Subspecies and Origin		IMMDL			TAMU			ICCDRU				
	N	BW	TW	Inc.	BW	TW	Inc.	BW	TW	Inc.	PCR Success	Accuracy
*A. g. grandis*												
Mexico												
Chihuahua	30	25	0	5	27	0	3	24	0	5	88.00%	100%
Coahuila	7	5	0	2	7	0	0	5	0	2	79.50%	100%
Durango	7	6	0	1	7	0	0	5	0	0	91.30%	100%
Tamaulipas	10	10	0	0	10	0	0	10	0	0	95.70%	100%
USA												
Mississippi	5	3	0	2	4	0	1	3	0	1	68.80%	100%
New, Mexico	1	1	0	0	1	0	0	1	0	0	85.70%	100%
Texas	17	12	0	5	14	0	3	3	2	5	74.30%	95.50%
*A. g. thurberiae*												
Mexico												
Chihuahua	17	0	14	3	0	16	1	1	14	2	90.80%	98.00%
USA												
Arizona	35	0	28	7	0	33	2	0	28	7	90.30%	100%

**Table 8 insects-14-00845-t008:** Accuracy of the ANZTMGE assay at each geographic location using a subset of weevil samples obtained from the Barr et al. (2013) study. From left to right, the columns correspond to the quantity of specimens per collection (N), weevil identity determinations (BW = boll weevil, TW = thurberia weevil, Inc. = inconclusive) made at each facility (IMMDL, TAMU, and ICCDRU) using the stepwise criteria (Figure 2), and the accuracy of the ANZTMGE assay for each collection.

Subspecies and Origin		IMMDL			TAMU			ICCDRU			
	N	BW	TW	Inc.	BW	TW	Inc.	BW	TW	Inc.	Accuracy
*A. g. grandis* (BW)											
Mexico											
Chihuahua, Delicias	13	9	0	4	10	0	3	10	0	3	100%
Chihuahua, Durango, Torreon	5	5	0	0	5	0	0	4	0	0	100%
Chihuahua, Ojinaga	8	8	0	0	8	0	0	7	0	0	100%
Chihuahua, Los Alamos	4	3	0	1	4	0	0	3	0	1	100%
Coahuila, Ejido Chavez	5	3	0	2	5	0	0	3	0	0	100%
Coahuila, San Pedro	2	2	0	0	2	0	0	1	0	0	100%
Durango, Jimenez	7	6	0	1	7	0	0	6	0	0	100%
Tamaulipas, Valle Hermoso	10	10	0	0	10	0	0	10	0	0	100%
USA											
Mississippi, Meyersville	5	3	0	2	4	0	1	3	0	1	100%
New Mexico, Artesia	1	1	0	0	1	0	0	1	0	0	100%
Texas, Rio Grande Valley	17	12	0	5	14	0	3	5	2	3	95.50%
*A. g. thurberiae* (TW)											
Mexico											
Chihuahua, Agua Prieta	17	14	0	3	16	0	1	14	1	2	98.00%
USA											
Arizona, Bisbee	1	1	0	0	1	0	0	1	0	0	100%
Arizona, East of Sasabe	12	12	0	0	12	0	0	12	0	0	100%
Arizona, Kitt Peak	9	4	0	5	7	0	2	6	0	3	100%
Arizona, Phoenix	5	4	0	1	5	0	0	3	0	2	100%
Arizona, Sonoita	4	4	0	0	4	0	0	2	0	0	100%
Arizona, Chiricahua, Mts.	4	3	0	1	4	0	0	4	0	0	100%

## Data Availability

The preliminary genome assembly used for read-mapping can be made available upon request to the authors. Reads from Raszick et al. [8] are available from the NCBI Sequence Read Archive (SRA) under BioProject ID PRJNA623635 and reads from Raszick et al. [21] are available from the NCBI SRA under BioProject ID PRJNA886713.

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
