# Peer review of "A New qPCR Assay for the Rapid Diagnosis of Anthonomus grandis Subspecies"

_insects, 2023, doi:10.3390/insects14110845_

Round 1

Reviewer 1 Report

Comments and Suggestions for Authors

The present manuscript provides a new method for the fast and accurate identification by qPCR of the pest species Anthonomus grandis sp.

Nowadays, in a global change world it becomes mandatory to perfom the most accurate pest identification in order to erradicate or control its populations with non invasive methods or limiting the use of pesticides.

Introduction:

This section is very complete and provides relevant information about previous works and protocols by PCR to identify the pest. However I feel that some parts sounds like a discussion.  For instance, in line 75-77 where is highlighted the time consuming with the COI diagnosis.

Materials and Methods:

I would like to know if the DNA extracted is evaluated in agarose geles (integrity) and using nanodrop (concentration) in order to discard false negatives due to degraded DNA.

I think is not necessary to provide the DNA volumes in PCR, just use the concentration, also with the MasterMix, you provide the final volume and its concentration.

Figure2: To me, It is a little bit difficult to read the blue squares with the white letters (in the printed version). It is a personal opinion but I suggest to change the colours.

In some parts you put the refferences in text using "Perkin et al. 2021" and in others the number, please review the manuscript.

The rest of the manuscript is completed and also well explained. Nothwithstanding I recommend to increase the number of citations to improve the discussion. 

Reviewer 2 Report

Comments and Suggestions for Authors

A very fast and simple protocol to identify and differentiate cotton boll weevil from its main subspecies (thurberiae weevil) is described. I guess the work is interesting and deserves to be published, after considering the following comments.

In the Introduction:

The reasons to develop a differentiation method for the two subspecies of Anthonomus grandis is well described in the introduction. However, I miss some information regarding the prevalence of each sub-species in the described interceptions at ports of entry (lines 66-67) or in field screenings, for example, when using pheromone traps in USA cotton fields.

Line 52: having different Gossypium species in the región that can be natural hosts of this pest and close relatives such as Turberia weevil, why focusing the presentation on their eradication?

Having a closely reated subspecies of Anthonomus grandis grandis, is there any chances to have interspecific (or sub-specific) hybrids, that could interfere with the molecular diagnostics of the target pest species boll weevil?

And related to this, which of the closer species of curculionids are sympatric to A. g. grandis?

In Methods:

Line 182: the boll weevils used in the evaluation of the diagnostic protocol were identified by “recognized experts”; even when there is no reason to have any doubt about this, a more objective criteria for classifying them must be presented or referenced.

Lines 196-7: please give more details (or a reference) with details of the co-precipitation with RNA. What is the source or RNA, quantities, etc? Did you discard it before measuring DNA concentration with the Qubit?

Regarding the Discussion

Line 500: it is stated that a cost-effective method for differentiating A. grandis subspecies has been described and validated, but I can not find any calculations to support this. If I am right, authors might add some comments on this, as well as costs figures, as supplementary material.
